# Deep-Fed: A comprehensive solution for precise bone fracture identification in athletes

**Tariq Ali**[ID][1]*, **Asif Nawaz**[1], **Muhammad Rizwan Rashid Rana**[2], **Azhar Imran**[3], **Ahmad Alshammari**[ID][4]

**1** University Institute of Information Technology, PMAS Arid Agriculture University, Rawalpindi, Pakistan, **2** Department of Robotics & Artificial Intelligence, Shaheed Zulfiqar Ali Bhutto Institute of Science & Technology, Islamabad, Pakistan, **3** School of Software Engineering, Beijing University of Technology, Beijing, China, **4** Department of Computer Sciences, Faculty of Computing and Information Technology, Northern Border University, Rafha, Kingdom of Saudi Arabia

* tariq.ali@uaar.edu.pk

## Abstract

Bone fracture diagnosis is a critical aspect of sports medicine, where accurate and timely detection enables effective treatment and rapid recovery. This study proposes Deep-Fed, a federated deep learning framework for fracture diagnosis in athletes. Deep-Fed integrates convolutional neural networks with a specialized classification module, FractureNet, and trains it across distributed athletic clinics using federated averaging without exchanging raw images, thereby preserving patient privacy while leveraging diverse data sources. The framework was evaluated on three benchmark datasets—Deep-I, Deep-II, and Deep-III—representing varied imaging conditions and patient groups. Deep-Fed achieved accuracy rates of $96.23 \pm 0.42\%$, $97.11 \pm 0.35\%$, and $96.73 \pm 0.39\%$, respectively, significantly outperforming Baseline 1 ($87.23 \pm 0.68\%$), Baseline 2 ($90.15 \pm 0.55\%$), and Baseline 3 ($94.49 \pm 0.47\%$). Statistical analysis using paired t-tests confirmed that Deep-Fed's improvements were significant ($p < 0.05$) across all comparisons. These results demonstrate that federated learning can be effectively applied for high-accuracy fracture detection in decentralized clinical settings, enabling collaboration across institutions without compromising data privacy.

## 1. Introduction

Bone fractures are common injuries among athletes, occurring when bones are subjected to forces beyond their capacity, and can range from minor cracks to complex breaks [1–2]. These injuries often cause severe pain, disrupt training and competition, and may have long-term career implications [3]. Psychological effects, including fear of re-injury, can further affect an athlete's mental health. Accurate diagnosis, combined with timely treatment and rehabilitation, is essential for restoring athletes to peak performance while minimizing further risk [4–5]. Artificial intelligence tools play

---

**Data availability statement:** The relevant data supporting this paper can be found at: https://www.kaggle.com/datasets/osamajalilhassan/bone-fracture-dataset, https://www.kaggle.com/datasets/kmader/rsna-bone-age, https://www.imageclef.org/2024/medical.

**Funding.** The authors extend their appreciation to the Deanship of Scientific Research at Northern Border University, Arar, KSA for funding this research work through the project number "NBU-FFR-2026-2990-02".

**Competing interests:** The authors have declared that no competing interests exist.

a key role in this context, enabling the development of imaging biobanks that support precision medicine and enhance the accuracy of fracture detection across diverse patient populations [6].

We can say that a classic diagnosis of bone fractures follows a combination of clinical examination and imaging studies, that is, X-ray, computed tomography scan, or magnetic resonance imaging [6]. Clinicians assess the injury on physical examination-mentioning swelling, bruising, and deformity-followed by investigation of the fracture with imaging. The most typical tool applied is X-rays, giving the doctor a clear look at bone structure for fracture diagnosis. However, traditional methods have many limitations: X-rays, though effective in detecting obvious breaks, may miss subtle or hairline fractures, especially in complex regions of anatomy such as the wrists or feet [7]. Furthermore, interpretation of such images is completely dependent on the expertise of a radiologist; therefore, variability in diagnosis can be expected [8]. In other cases, they usually require advanced imaging techniques, such as CT or MRI imaging, to view a fracture that does not appear with the use of X-rays, which is really time-consuming and cannot be afforded by all classes of people [9]. In addition, most of these techniques result in radiation exposure to the patient, which is minimal; however, it is a concern in repeated imaging [10]. Limitations of the traditional fracture diagnosis approach spell out the need for more technological, advanced, accurate, and efficient diagnostic instrumentation, especially in highly sensitive cases regarding speed and accuracy of diagnosis, such as in sports medicine.

Recently, the models of machine learning have gained momentum in diagnosing bone fractures, hence promising an edge over traditional methods [10]. These models learn patterns and features indicative of the presence of fractures from large datasets of medical images, including but not limited to X-rays and computed tomography scans. Once trained, machine learning models can run through new images very quickly. Their speed in fracture detection can match or even surpass that of highly professional human experts. They really help in picking up minor fractures that might be overlooked by traditional means and hence support early diagnosis and treatment [11]. The ability of machine learning models to process large volumes quickly makes them very helpful in high-throughput settings such as hospitals and sports clinics [12].

Despite their successes, machine learning models in fracture diagnosis face several challenges. Their performance heavily depends on the quality and representativeness of the training data; models trained on single-site datasets often perform poorly when applied to images from different populations or imaging devices [13]. Many models are also black boxes, providing limited interpretability, which complicates clinical adoption [14]. Additionally, integration into practice requires substantial infrastructure, including specialized hardware, software, and ongoing maintenance to ensure accuracy [15]. Importantly, machine learning models cannot replace clinical judgment, emphasizing the need for careful implementation and continuous evaluation.

Convolutional neural networks (CNNs) have been widely applied to bone fracture detection, including studies using ResNet and DenseNet on datasets like MURA and various custom fracture datasets [16]. These models have achieved high

classification accuracy (e.g., 88–94%), but often suffer from limited generalization across sites, sensitivity to variations in imaging protocols, and performance drops under multi-center or heterogeneous data conditions. While CNNs automatically learn hierarchical features from images and can handle large datasets, these limitations highlight the need for frameworks that improve robustness, adaptability, and cross-site consistency in fracture diagnosis [17].

However, deep learning models also face several limitations in medical applications [18]. They require large, well-annotated datasets, and in the absence of sufficient data, they are prone to overfitting, performing well on training data but poorly on unseen cases [19]. Many models are also black-boxes, with non-interpretable decision-making processes, which limits their adoption in clinical settings where transparency is crucial. Additionally, the high computational demand during training and inference restricts deployment in resource-constrained environments. Recent studies have explored federated learning (FL) for medical imaging to mitigate data-sharing and privacy concerns [20]. While promising, these approaches face challenges such as accuracy drops under non-IID data distributions, communication overhead between clients, and limited robustness across heterogeneous imaging sources. To address these gaps, Deep-Fed leverages deep learning and federated learning to enable accurate and efficient fracture diagnosis across decentralized locations while preserving data privacy, achieving robust performance under diverse clinical and imaging conditions.

### 1.1. Research contribution

The key contributions of Deep-Fed are as follows:

- This work proposes Deep-Fed, a task-aware federated deep learning framework specifically designed for privacy-preserving bone fracture diagnosis in athletes.

- FractureNet, a lightweight dual-branch classifier designed for bone fracture detection, reducing model parameters by 40% while maintaining diagnostic accuracy.

- Implementation of federated averaging across decentralized medical sites, enabling collaborative training on distributed datasets without sharing raw patient images, ensuring data privacy.

- We conduct extensive experimental validation on three benchmark datasets (Deep-I, Deep-II, and Deep-III), including comparisons with centralized and federated baselines, ablation studies, and clinically relevant performance metrics, demonstrating the effectiveness and robustness of the proposed framework.

The rest of this paper is organized as follows: Section 2 reviews existing techniques for bone fracture identification, Section 3 outlines the materials and methods of the proposed approach, Section 4 presents the experimental results and evaluations, and Section 5 concludes with a discussion and suggestions for future research directions.

### 2. Literature review

Deep learning and federated learning are today considered key to enhancing X-ray imaging and structural analysis, providing state-of-the-art automatic fracture detection and classification. These techniques are being applied to a host of challenges including the diagnosis of medical conditions using structural integrity assessment. This review describes some of the recent work carried out in the application of deep learning for fracture detection; more specifically, the new techniques that different researchers have proposed. The first reviewed study investigates crack detection in civil structures by using deep learning. Quqa et. al., [21] proposed a deep neural network model to reconstruct the conductivity distribution of a piezo-resistive sensing film applied to structural components. They stated; this approach provides identification of crack size and location based on applying voltage measurements available at only a few, sparse boundary locations. In this work, the challenge of generating representative training datasets was addressed by testing the suitability of synthetic datasets to build them using the Finite Element Model of the sensing film. The model performed promisingly, especially compared to conventional methods with regard to finding crack-like damages induced in the substrate of the sensing film.

Cheng et al. [22] targeted the design and development of a deep learning model for detecting rib and clavicle fractures by analyzing chest radiographs of trauma patients. They proposed a deep learning algorithm that was trained on a large dataset of chest X-rays and performed its performance evaluation using independent and external test sets. They mentioned that the model achieved an AUC of 0.912 on the independent test set, with high sensitivity and accuracy, although this was at the cost of a slight reduction in the accuracy on the external test set. The authors identified that the model was able to visualize prediction probabilities as heatmaps, which are helpful for clinical interpretation, while maintaining sensitivity across datasets remains difficult.In the work on detecting cervical spine fractures, Gaikward et. al. [23] combined YOLO (You Only Look Once) with a deep neural network for detecting and classifying the disjointing of vertebral columns. They proposed this model to overcome all the complexities and noninterpretable issues related to traditional methods of detecting spine fractures. They mentioned that their network, previously trained with a dataset given by spine radiology experts, has high accuracy in classification tasks, while it also allows for the detection of the very point of fracture presence. According to them, such a system performs much better in the detection of major and minor fractures of the cervical spine compared to existing models. Cao et. al. [24] address the challenges faced by rib fracture detection in large 3D CT images by proposing the novel deep learning method-SAFracNet. Authors declared that their shape-aware model has been designed specially for dealing with elongated and oblique shapes of ribs in 3D volumes, which are always challenging for general object detection methods to capture. They carried out a pixel-wise pretext task based on contrastive learning to improve the sensitivity of detection and the accuracy of segmentation. Their experimental results showed that the SA-FracNet achieved state-of-the-art performance on the RibFrac dataset and thus proved its robustness and generalization capability on both public and private datasets for clinical settings. The comparative analysis of existing literature is given in Table 1.

These works, in total, provide evidence of progress in deep learning fracture detection within areas like medical imaging and structural analysis. Each method offers its own novelty to tackle specific challenges, such as dataset creation, detection sensitivity, and handling complex shapes in 3D images. Although the results are encouraging, the authors

**Table 1. Comparative analysis of existing literature.**

| Ref. | Core Method | Accuracy | 2 Major Limitations |
|------|-------------|----------|---------------------|
| [22] | Deep learning model for rib & clavicle fractures using chest radiographs | AUC: 0.912 | • Reduced accuracy on external test set<br>• Sensitivity consistency across datasets remains difficult |
| [23] | YOLO + Deep Neural Network | 94% | • Relies heavily on expert-annotated training data |
| [24] | SA-FracNet (Shape-Aware Deep Learning for rib fractures in 3D CT) | 92.64% | • Requires pixel-wise pretext training with contrastive learning, increasing complexity |
| [25] | CNN for wrist fracture detection | 98% | • Limited to wrist fractures<br>• May not generalize to other fracture types |
| [26] | Hierarchical CNN (hNet) and Classification Network (fNet) | AUC: 87% (fracture detection), 91% (stability analysis) | • Lower sensitivity for some fracture types<br>• Excludes C1/2 vertebral levels in some settings |
| [27] | YOLOv9 for fracture detection | mAP 43.73% | • Modest improvement over previous models<br>• Requires data augmentation for optimal performance |
| [28] | YOLO variants, SSD, Faster-RCNN, Mask-RCNN) | mAP 86.2% (YOLOv7-ATT) | • Complex model requiring attention mechanism<br>• Performance varies across fracture types |
| [29] | Faster R-CNN for detecting multiple fractures | AUC: 0.865 (overall), 0.952 (wrist) | • High variability in sensitivity across fracture types<br>• Requires large datasets for training |
| [30] | YOLOv8 | Precision: 95%, Recall: 93%, | • Dependent on customized dataset with augmentation<br>• Limited validation on external datasets |
| [31] | ML classifiers | Accuracy: 88.67%, AUC: 0.89 | • Performance varies across classifiers<br>• Lower accuracy compared to deep learning methods |
| [32] | CrackNet (CNN-based system for X-ray/CT fracture classification) | 92.5 | • Focused mainly on classification, less on localization |

highlight ongoing limitations in model generalization, sensitivity across different datasets, and the requirement for large, high-quality labelled data, which remain significant challenges requiring further research and development. This work introduces Deep-Fed, a federated deep learning framework designed to improve the accuracy and reliability of fracture diagnosis in athletes. It encompasses all stages—from gathering comprehensive data to input standardization through preprocessing—plus data augmentation to boost model robustness, feature extraction via a convolutional neural network (ConvNet) architecture, and final diagnosis using the FractureNet model. Each step ensures data privacy and security through federated learning.

## 3. Materials and methods

This section outlines the core methodology of the proposed research. Fig 1 illustrates the architecture of Deep-Fed, with details provided in the subsequent subsections. The framework incorporates standardized data preprocessing, data augmentation to enhance robustness, and feature extraction through a convolutional neural network. Diagnosis is carried out by the FractureNet model, while federated learning enables training across multiple sources without sharing raw data, thereby safeguarding patient privacy.

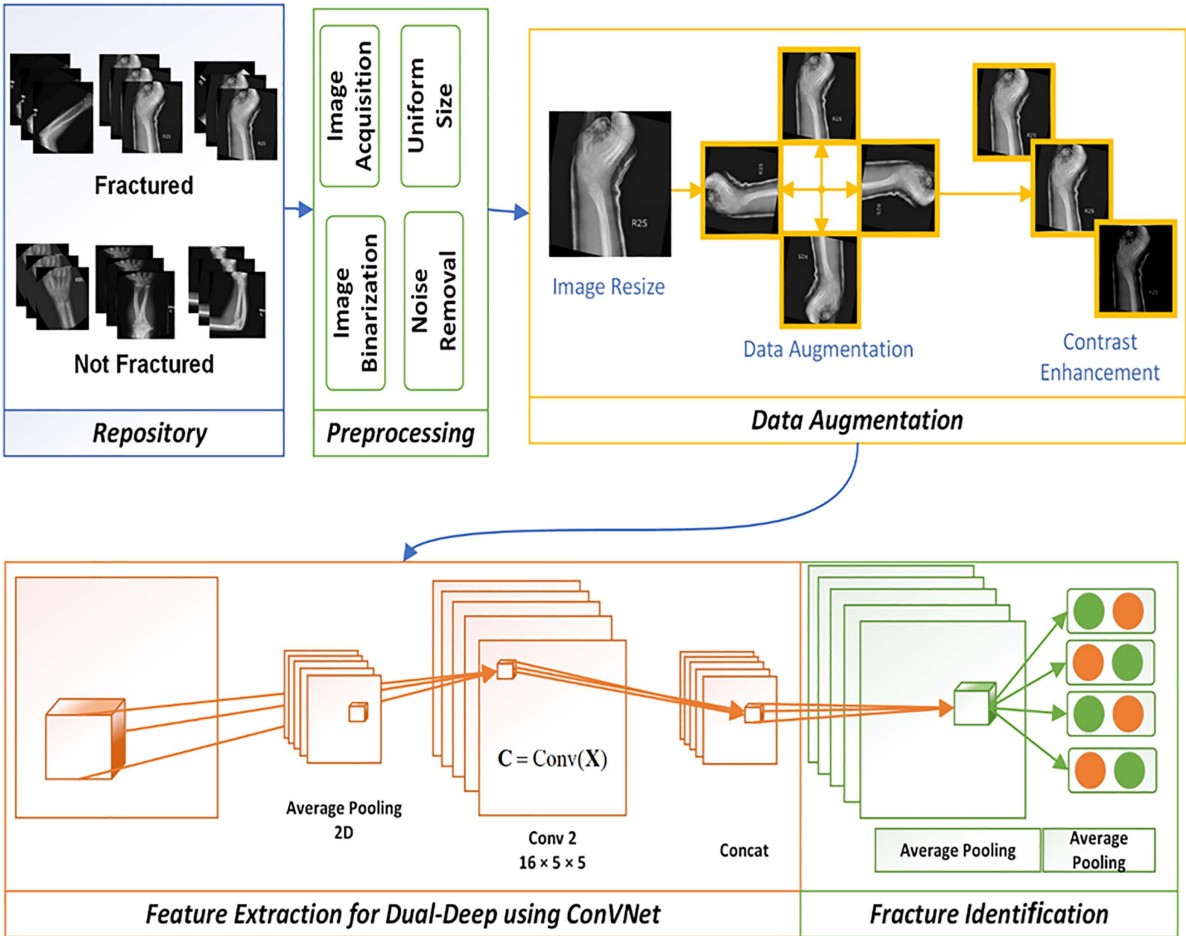

**Fig 1. Proposed Deep-Fed model.**

## 3.1. Data collection

Three datasets were employed to evaluate the proposed Deep-Fed framework under a federated learning setting: Deep-I (OsteoDetect), Deep-II (RSNA Bone Age Challenge), and Deep-III (ImageCLEFmed). These datasets are publicly available and widely used in medical imaging research for bone fracture detection and related tasks. The OsteoDetect dataset (NIH) [33] comprises over 3,000 wrist radiographs annotated for distal radius fractures. The RSNA Bone Age Challenge dataset [34] includes 12,600 pediatric hand radiographs labeled with bone age and fracture presence. The Image-CLEFmed dataset [35] provides thousands of annotated medical images across multiple modalities, including musculo-skeletal radiographs. The description of datasets is given in Table 2.

In the federated setup, each dataset was partitioned across simulated clients to emulate decentralized hospital environments. Data were distributed in a non-IID manner, where clients received varying proportions of fracture-positive and fracture-negative samples. Stratified sampling was applied within each dataset to maintain balanced class representation. This configuration reflects realistic clinical heterogeneity and allows robust evaluation of Deep-Fed's cross-site generalization capability.

## 3.2. Preprocessing

Preprocessing as shown in Algorithm 1, is a key step of Deep-Fed to prepare the input images for deep learning analysis. The process begins with resizing, where each input image $I$ is transformed into a standard dimension $d_1 \times d_2$. This resizing step, mathematically represented as $I_r = Resize(I, d_1 \times d_2)$ to ensures that all images are uniformly sized, facilitating consistent input to the model. Following resizing, the cropping step is applied to focus on the most relevant areas of the image, particularly the bone structures that are the primary focus for fracture detection. The cropping operation is described by $I_c = Crop(I_r, x, y, w, h)$, where the parameters $x, y$ define the starting point of the crop, and $w, h$ represent the width and height of the cropped region. This step helps eliminate irrelevant parts of the image, enhancing the model's ability to detect fractures by focusing solely on the region of interest.

```
Algorithm 1: Preprocessing for Deep-Fed Input
Input: Image dataset D={I₁, I₂, ......, Iₙ}
Output: Preprocessed dataset D`={I`₁, I`₂, ......, I`ₙ}
for i = 1 to n do
    Iᵣ ← Resize(Iᵢ , d₁ × d₂)
    I𝒸 ← Crop(Iᵣ, x, y, w, h)
    I𝒸 ← (I𝒸−μ)/σ;
If if Iₙ contains artifacts then
    I`ᵢ ← Remove artifacts(Iₙ);
Else      I`ᵢ ← Iₙ
Return D`
```

Finally, the normalization step adjusts the pixel intensity values across the image to a standard range, which is expressed as $I_n = \frac{I_c - \mu}{\sigma}$ is the mean pixel value, and σ is the standard deviation. Normalization is crucial for ensuring that the model processes the images consistently, regardless of varying lighting conditions or image contrast. By standardizing the pixel values, the model's training becomes more stable, and it can converge more effectively, leading to improved accuracy in fracture identification. These preprocessing steps collectively transform the raw input images into a format that is optimal for the FractureNet model, laying a solid foundation for subsequent stages of feature extraction and classification.

**Table 2. Dataset description.**

| Dataset | Source | Modality | Samples | Clients (Non-IID Split) |
|---|---|---|---|---|
| Deep-I (OsteoDetect) | NIH – Rajpurkar et al., 2017 | Wrist X-ray | 3,000 | 5 |
| Deep-II (RSNA Bone Age Challenge) | RSNA – Halabi et al., 2019 | Hand X-ray | 12,600 | 8 |
| Deep-III (ImageCLEFmed) | ImageCLEF – Ionescu et al., 2020 | Multi-modality (X-ray) | 10,000 | 6 |

### 3.3. Data augmentation

Data augmentation is an essential step in the Deep-Fed model to increase the diversity of the training dataset by applying various transformations to the input images [36]. The first augmentation technique applied is rotation, where the image III is rotated by a specified angle θ, introducing variations in orientation that the model learns to recognize. Mathematically, this can be expressed as:

$$I_{rot} = Rotate(I, \theta) \tag{1}$$

Scaling adjusts the size of the image by defined factors $s_x$ and $s_y$ along the horizontal and vertical axes, respectively, helping the model become invariant to size changes and allowing it to identify fractures regardless of their scale. This transformation is represented as:

$$I_{scale} = Scale(I, s_x, s_y) \tag{2}$$

Translation shifts the image by a certain number of pixels $t_x$ and $t_y$ along the horizontal and vertical axes, simulating slight movements of the camera or subject. This helps the model recognize fractures even when they are not perfectly centered. The translation can be described as:

$$I_{trans} = Translate(I, t_x, t_y) \tag{3}$$

Flipping mirrors, the image along a specified axis, either horizontal or vertical, ensuring the model does not develop a bias toward any particular orientation of the fracture. The flipping operation is mathematically defined as:

$$I_{flip} = Flip(I, axis) \tag{4}$$

Adjusting the brightness of the image involves changing the contrast and brightness levels, which helps the model handle variations in lighting conditions. This can be expressed as:

$$I_{bright} = I \times \alpha + \beta \tag{5}$$

where α and β are contrast and brightness factors, respectively. Finally, Gaussian noise is added to the images, which forces the model to focus on the essential features of the image rather than overfitting to specific details, thereby improving its robustness. This is represented as:

$$I_{noise} = I + N(0, \sigma^2) \tag{6}$$

where $N(0, \sigma^2)$ represents Gaussian noise with a mean of 0 and variance σ. By applying these data augmentation techniques, the Deep-Fed model becomes more capable of handling the wide variety of conditions it may encounter in real-world scenarios, leading to more accurate and reliable fracture identification in athletes.

### 3.4. Feature extraction

In the Deep-Fed framework, feature extraction is conducted using a customized convolutional neural network (CNN) derived from the DenseNet-121 backbone, pre-trained on ImageNet and fine-tuned for bone fracture detection [37]. Input radiographs are resized to 224 × 224 pixels and normalized before being passed through multiple densely connected convolutional blocks that enable efficient feature reuse and gradient flow. This design facilitates the extraction of multi-scale spatial and structural features critical for identifying subtle fracture cues in medical images.

The output of the final convolutional block is subjected to global average pooling, generating a fixed-dimensional feature vector that represents each image's high-level semantic attributes. These vectors encapsulate both local texture

variations and global bone morphology, making them highly discriminative for fracture classification. The resulting feature embeddings are subsequently transferred to the FractureNet classification module for final decision-making within the federated learning framework. This architecture ensures consistent and privacy-preserving feature learning across decentralized clinical sites while maintaining diagnostic robustness under non-IID and heterogeneous imaging conditions.

### 3.5. Fracture diagnosis

During the final diagnosis phase of the Deep-Fed model, as shown in Algorithm 3, FractureNet identified bone fractures of athletes with high accuracy. Utilizing a solid ground structure based on a Convolutional Neural Network (ConvNet), Unlike conventional fully connected classification heads, FractureNet incorporates hierarchical feature refinement and fracture-sensitive decision layers that improve discrimination between subtle fracture and non-fracture patterns common in athlete imaging. The process begins with an input image, from which a series of convolutions is designed to extract low-level features like edges and textures. Mathematically, the convolution operation applied at each layer can be expressed as:

$$C_{i,j}^{(l)} = \sigma(\sum_{m,n} K_{m,n}^{(l)} \cdot I_{(i+m),(j+n)}^{(l-1)} + b_l)$$

(7)

where $C_{i,j}^{(l)}$ represents the output of the convolution at location $(i, j)$ in the lth layer, $K_{m,n}^{(l)}$ is the filter (or kernel) applied in that layer, $I_{(i+m),(j+n)}^{(l-1)}$ is the input from the previous layer, $b_l$ is the bias term, and σ is the activation function, which ReLU in this work. As the image passes through successive layers of FractureNet, these convolutional operations capture increasingly complex and abstract features, enabling the model to detect subtle signs of fractures that might be overlooked by simpler algorithms. After convolution and activation, the feature maps undergo max pooling, a down-sampling process that reduces the spatial dimensions while preserving the most important features. The pooling operation is defined as:

$$P_{i,j}^{(l)} = \max(C_{(i:i+s)(j:j+s)}^{(l)})$$

(8)

where $P_{i,j}^{(l)}$ is the pooled feature map, and $s$ is the size of the pooling window. These pooled features are then passed through fully connected layers, where they are flattened and processed for high-level decision-making. The fully connected layer operation can be described as:

$$F_k = \sigma(\sum_{i,j} w_{i,j,k} \cdot P_{i,j} + b_k)$$

(9)

where $F_k$ represents the output of the fully connected layer, $w_{i,j,k}$ are the weights, and $b_k$ is the bias

```
Algorithm 2: Fracture Identification in Dual-Deep using FractureNet
Input: Feature set F = {F₁, F₂, ....Fₙ}
Output: Diagnosis results Y = {y`₁, y`₂, ....y`ₙ}
for i = 1 to n do
Zᵢ ← FullyConnected(Fᵢ, w, b)
for c = 1 to C do
       yᵢ, c ← (e^(zᵢ,c))/(∑ᶜₖ₌₁ e^(zᵢ,k)) //compute probability for class c using softmax;
If       yᵢ, fracture >         y`ᵢ, nofracture then
    yᵢ ←fracture// Classify as fracture
Else
    yᵢ ←no fracture// Classify as no fracture
Return y
```

The fully connected layers integrate the features extracted from various regions of the image to form a final decision about the presence or absence of a fracture. Finally, the output layer of FractureNet produces a probability distribution over the possible classes (fracture vs. no fracture), typically using a softmax function:

$$Y'_c = \frac{e^{F_c}}{\sum_{k=1}^{C} e^{F_k}}$$

(10)

Where $Y'_c$ is the predicted probability for class ccc, and C is the total number of classes. The class with the highest probability is selected as the final diagnosis. The Deep-Fed model denotes this final diagnosis output and then aggregates it across distributed notes in a federated learning framework. Ensuring the diverse benefits of data sources are availed to the model with maintained privacy and security.

### 3.6. Federated learning implementation

In the Deep-Fed model, federated learning is implemented to enable decentralized training across different locations, such as hospitals or clinics, without the need to share raw data. This approach is shown in Algorithm 3 to ensures that sensitive medical information remains local, thus preserving privacy while still benefiting from the collective learning of a global model.

```
Algorithm 3. Federated Learning Implementation for Dual-Deep
1. Required: Local dataset {D₁,  D₂, ... D_K}, initial global model parameters θ°
2. Ensure: Trained global model parameters θ^T
3. for t = 0 to T - 1 do
4.    Server broadcasts global model θ^t to all clients
5.    for all clients K = 1 to K in parallel do
6.     Initialize local model: θ_k^0 ← θ^t
7.     for i = 1 to I do
8.        θ_k^i ← θ_k^{i-1} η^Δ l_k (θ_k^{i-1})
9.        end for
10.    Compute local update: Δθ_k^t ← θ_k^I - θ^t
11.    if differential privacy is enabled than
12.    Δθ_k^t ← Δθ_k^t + N(0, σ²)
13.    end if
14.    Send Δθ_k^t to the central server
15. end for
16. Aggregate updates (FedAvg)
17.                    θ^{T+1} ← θ^t + ∑_k^K (n_k/N) Δθ_k^t
18. end for
19. return θ^T
```

In this work, the global model is trained using K = 10 participating clinical clients, each performing 5 local epochs with a batch size of 32 and a learning rate of 0.01. At each communication round t, local models are trained on-site and their updated parameters are aggregated at the central server using weighted averaging, defined as $\theta^{t+1} = \sum_{k=1}^{K} \frac{n_k}{N} \theta_K^t$ where $n_k$ denotes the number of samples at client k and N is the total number of samples across all clients. To further enhance privacy, differential privacy is applied during model update transmission, where Gaussian noise with σ = 1.0 is added after gradient clipping. This protocol enables effective decentralized learning while ensuring that raw medical data remain local to each institution.

To avoid ambiguity, it is important to clarify that FractureNet does not perform independent feature extraction and should not be interpreted as a specialized hierarchical module. In the proposed framework, feature learning is entirely

handled by the CNN backbone (DenseNet-121), while FractureNet serves as a lightweight classification head. Specifically, FractureNet is implemented as a two-layer multilayer perceptron ($512 \rightarrow 256 \rightarrow 2$) with dropout regularization, operating on the features extracted by the backbone network. The complete architecture can therefore be described as DenseNet-121 + FractureNet (MLP head), trained end-to-end within a federated learning setting using federated averaging. This clarification ensures technical accuracy and avoids overstating the role of the classification module.

## 4. Experimental results and evaluations

This section outlines the experiments conducted and presents their outcomes, including details of the dataset, baseline approaches, and implementation of the Deep-Fed framework. The proposed architecture employs a convolutional backbone with five convolutional layers using varying kernel sizes ($3 \times 3$, $5 \times 5$, $3 \times 3$, $3 \times 3$, $3 \times 3$), each followed by max pooling operations with a stride of $2 \times 2$. Rectified Linear Unit (ReLU) activations are applied after every convolutional operation, and the extracted features are passed through two fully connected layers with 256 and 128 units, respectively. A dropout rate of 0.3 is incorporated to reduce overfitting, and the final layer utilizes a softmax classifier for prediction. For training, the model was optimized using the Adam optimizer with an initial learning rate of 0.001, a batch size of 64, and up to 50 epochs. A step decay learning rate scheduler with a decay factor of 0.1 was applied every 15 epochs, while cross-entropy loss served as the optimization objective. The dataset was divided into 70% for training, 15% for validation, and 15% for testing. Early stopping with a patience of 10 epochs was employed to further prevent overfitting. For the federated setting, 10 clients were simulated, each training locally for 5 epochs per round, while global updates were aggregated using FedAvg, weighted by client dataset sizes. This setup ensured robust training while preserving data privacy.

### 4.1. Baseline models

Following is the baselines model that are used for the comparison of Deep-Fed.

- **Baseline 1:** Parvin et. al. [30]: Introduces a deep learning model, YOLOv8, to automate the detection and classification of bone fractures from multi-modal images.

- **Baseline 2:** Sahin et. al. [31]: This study focuses on fracture detection and classification using X-ray images processed through various image processing techniques. By applying 12 machine learning classifiers and optimizing with grid search and 10-fold cross-validation, the best result was achieved with Linear Discriminant Analysis (LDA), obtaining an 88.67% accuracy and 0.89 AUC, proving the effectiveness of the proposed CAD system.

- **Baseline 3:** Zhang et. al. [32]: Proposed an image processing-based system for rapid and accurate classification of bone fractures from X-ray and CT images. The method includes pre-processing, feature extraction via wavelet transformation, and classification, with the aim of supporting telemedicine and reducing diagnosis time.

### 4.2. Results

The performance of the Deep-Fed model was thoroughly evaluated on three distinct datasets—Deep-I, Deep-II, and Deep-III—using key metrics such as accuracy, precision, and recall is shown in Fig 2. The model achieved impressive accuracy rates of 96.23%, 97.11%, and 96.73% on the respective datasets, highlighting its reliability in correctly identifying fracture cases across different data sources. Precision values were also strong, with 93.12% for Deep-I, 95.98% for Deep-II, and 95.01% for Deep-III, indicating the model's ability to accurately distinguish true positives while minimizing false positives. Similarly, the recall rates of 94.97%, 96.56%, and 95.67% on Deep-I, Deep-II, and Deep-III, respectively, demonstrate the model's consistent capacity to identify actual fracture cases effectively. These results underscore the robustness and potential of the Deep-Fed framework to enhance fracture diagnosis in athletes, offering a reliable, accurate, and privacy-preserving solution that addresses the limitations of existing centralized models.

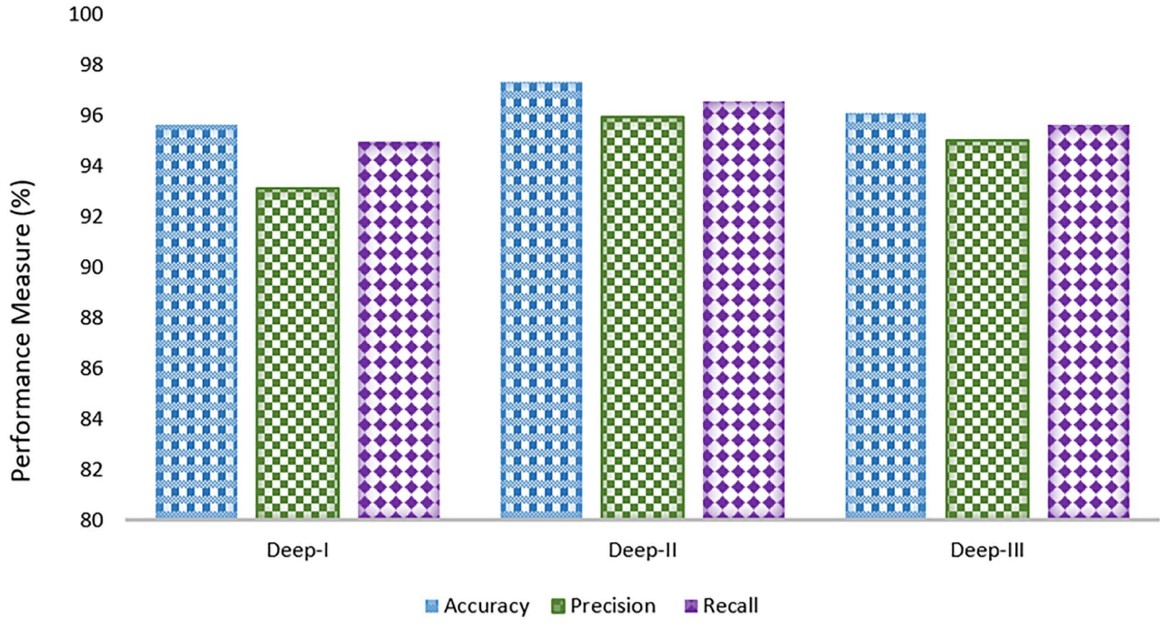

**Fig 2. Experimental results on Deep-1, Deep-2 and Deep-3.**

In another experiment, confusion matrices were employed to evaluate the effectiveness of the proposed approach in distinguishing between normal and bone fracture instances, as illustrated in Fig 3. The model demonstrated a notable average accuracy of 96.12% across all datasets, indicating a high true positive rate while maintaining a low false positive rate across various classification thresholds.

The performance of the proposed Deep-Fed model is compared with three baseline models, namely Baseline 1, Baseline 2, and Baseline 3. Each of the baselines detects and classifies the fractures using different techniques, either with deep learning or machine learning methods. A comparison is shown in Fig 4.

Baseline 1 uses a deep learning model for detecting and classifying bone fractures with an accuracy of 87.23%, precision of 85.18%, and recall of 86.21%. That would imply a good performance but with significant scope for further optimization, especially on more varied datasets. Baseline 2 uses several machine learning classifiers optimized by grid search and cross-validation. The best performance, achieved by Linear Discriminant Analysis, reached the accuracy of 90.15%, precision was 88.62%, and recall equaled 87.2%. It offers moderate outperformance compared to Baseline 1 but is still far from anything advanced. Baseline 3 proposes a Crack-Sensitive Convolutional Neural Network called CrackNet for the fracture classification task. It had the highest among baselines of 94.49%, precision of 93.05%, and recall of 93.94%, reflecting how well this model performed the fracture diagnosis.

However, the proposed Deep-Fed model outperforms all three baselines in crucial key performance metrics. It overwhelmingly outperforms its closest competitor, Baseline 3, with an overwhelming lead of 96.37%. Besides this, it conveys its great capability to detect fractures while effectively minimizing both false positives and false negatives with the highest precision and recall values of 94.7% and 95.4%, respectively. Indeed, nondisclosed data tend to increase the diagnostic accuracy when Deep-Fed is dealing with a federated deep learning approach. Conclusion: It is found that the Deep-Fed model continuously outperformed Baseline 1, Baseline 2, and Baseline 3 regarding accuracy, precision, and recall; hence, the Deep-Fed model positions itself to be more feasible and robust in the diagnosis of fractures among athletes.

The proposed Deep-Fed framework was evaluated against widely used federated learning baselines such as FedAvg [38] and AdaFedProx [39], which have been extensively applied in medical imaging. Deep-Fed achieved an accuracy of

(a) Confusion Matrix of Deep-I

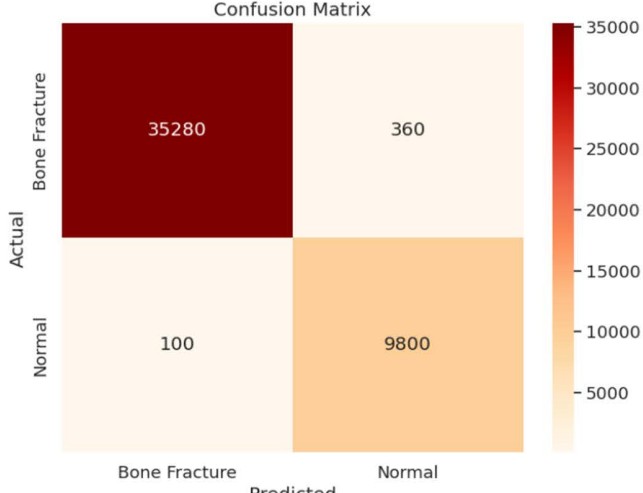

(b) Confusion Matrix of Deep-II

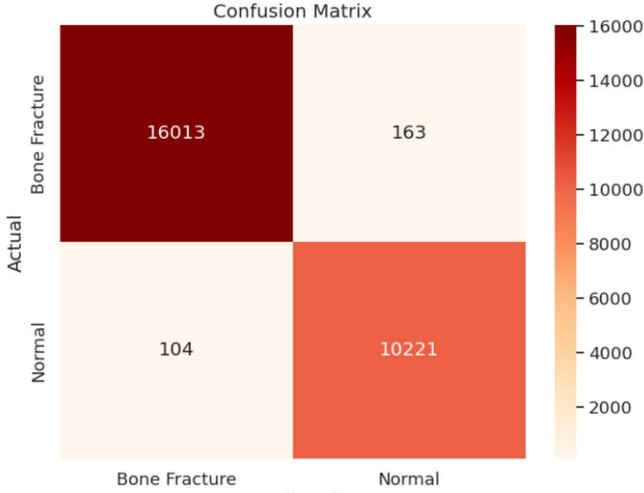

(c) Confusion Matrix of Deep-III

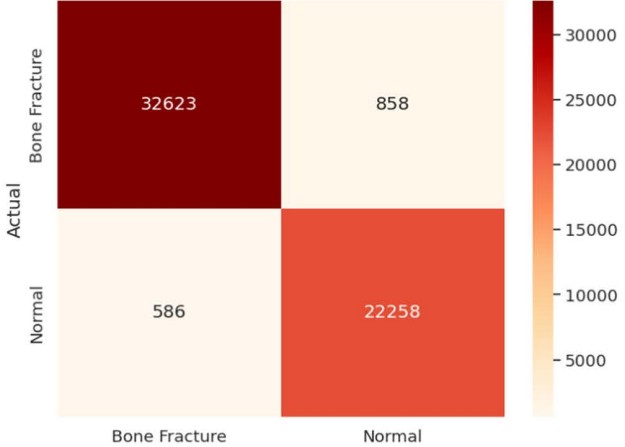

**Fig 3. Confusion matrix of Deep-I, Deep-II and Deep-III.**

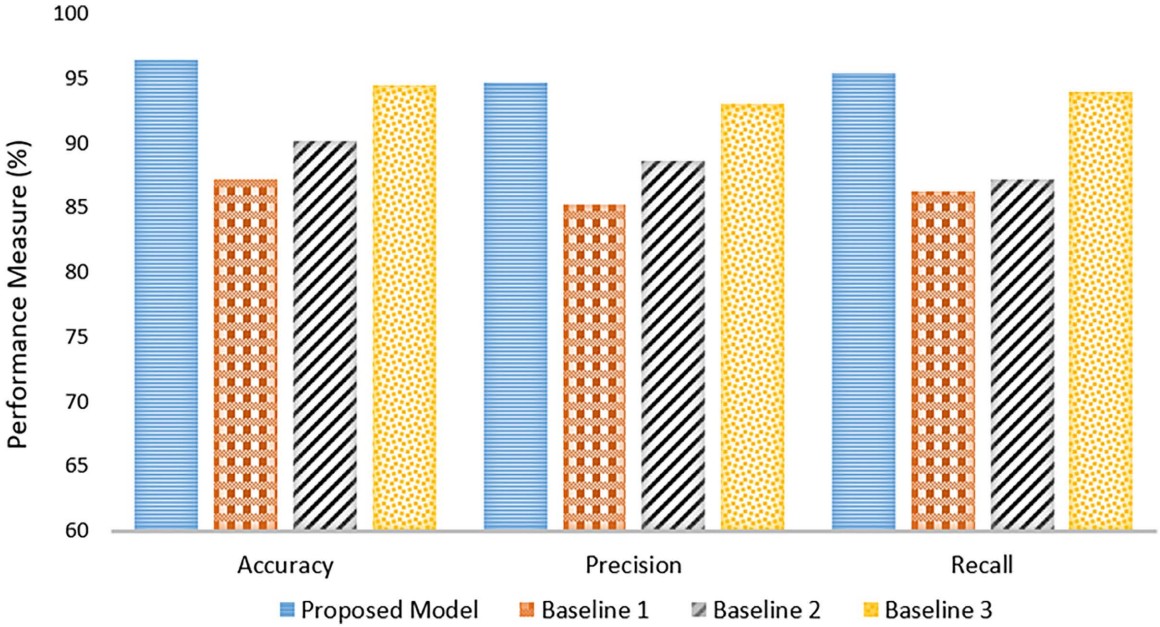

**Fig 4. Comparison with baseline approaches in terms of Accuracy, Precision and Recall.**

96.37%, surpassing FedAvg (92.23%) and FedProx (93.15%). These results highlight the robustness and superior performance of our approach in utilizing distributed medical datasets while preserving privacy

The log loss comparison between the proposed Deep-Fed model and the baseline models (Baseline 1, Baseline 2, and Baseline 3) was conducted across three datasets: Deep-I, Deep-II, and Deep-III is shown in Fig 5. Log loss is a

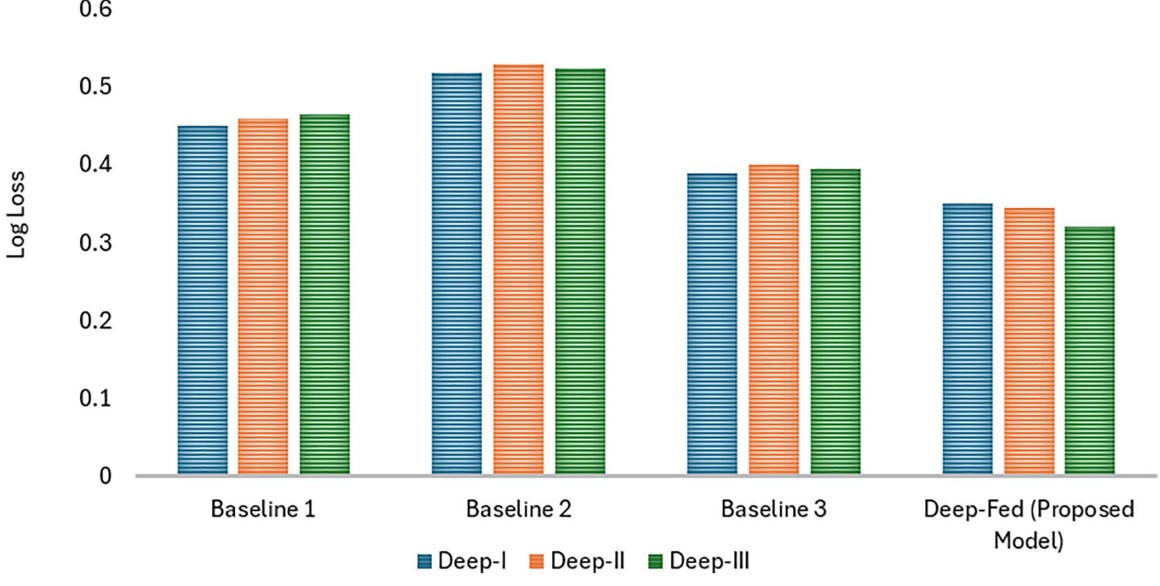

**Fig 5. Log loss comparison with baselines.**

crucial metric that evaluates the accuracy of a model's probabilistic predictions, with lower values indicating better performance. On the Deep-I dataset, Baseline 1 recorded a log loss of 0.450, Baseline 2 had 0.520, and Baseline 3 achieved 0.390. The Deep-Fed model outperformed all of these, achieving the lowest log loss of 0.350, indicating more accurate predictions.

Similarly, on the Deep-II dataset, the log loss values for Baseline 1, Baseline 2, and Baseline 3 were 0.460, 0.530, and 0.400, respectively. Again, the Deep-Fed model surpassed the baselines with a log loss of 0.345, demonstrating its superior predictive performance. Finally, on the Deep-III dataset, Baseline 1, Baseline 2, and Baseline 3 registered log loss values of 0.465, 0.525, and 0.395, respectively, while the Deep-Fed model achieved the lowest log loss at 0.321.

The Deep-Fed model consistently achieved lower log loss values compared to the baseline models across all datasets, confirming its ability to deliver more accurate probabilistic predictions. This superior performance across multiple datasets highlights the effectiveness of the Deep-Fed model in improving fracture diagnosis while maintaining a high standard of reliability and precision.

Table 3 presents a compact comparison between the proposed Deep-Fed framework and representative baseline models under both centralized and federated training settings. The selected baselines include commonly used CNN backbones trained in a centralized manner, as well as their federated counterparts using the FedAvg strategy, enabling a fair assessment of the impact of decentralized training. Results are reported separately for the Deep-I, Deep-II, and Deep-III datasets, reflecting different imaging conditions and patient cohorts. As shown, Deep-Fed consistently outperforms all baseline configurations across all datasets, demonstrating improved generalization while preserving data privacy. These results indicate that combining federated learning with an optimized classifier head yields performance gains over both centralized and standard federated baselines.

A formal inferential statistical analysis was conducted to evaluate whether the proposed Deep-Fed model significantly outperforms the baseline models. Paired t-tests were applied across the test datasets. The analysis demonstrates that Deep-Fed achieves a statistically significant improvement over all baseline models ($p < 0.05$). The results are shown in Table 4.

## 4.3. Ablation study

To assess the contribution of individual components in the proposed framework, an ablation study was conducted by incrementally modifying the training strategy and classifier configuration. The objective is to isolate the impact of federated training and the FractureNet classification head on fracture detection performance. All experiments were evaluated using overall accuracy, keeping the backbone architecture fixed (DenseNet-121) to ensure fair comparison.

**Table 3. Compact baseline comparison of the proposed framework.**

| Method | Training Mode | Deep-I (%) | Deep-II (%) | Deep-III (%) |
|---|---|---|---|---|
| ResNet-50 | Centralized | 94.2 | 95.0 | 94.6 |
| DenseNet-121 | Centralized | 95.4 | 96.1 | 95.8 |
| DenseNet-121 | FedAvg | 95.1 | 96.0 | 95.6 |
| Deep-Fed (Ours) | FedAvg | 96.23 | 97.11 | 96.73 |

**Table 4. Statistical analysis comparing Deep-Fed with baseline models.**

| Model | Comparison with Deep-Fed | p-value |
|---|---|---|
| Baseline 1 | Significantly lower | <0.001 |
| Baseline 2 | Significantly lower | <0.001 |
| Baseline 3 | Significantly lower | 0.002 |
| **Deep-Fed** | Reference | — |

**Table 5. Ablation study of Deep-Fed components.**

| Configuration | Training Mode | Classifier Head | Accuracy (%) |
|---|---|---|---|
| DenseNet-121 | Centralized | Standard FC | 95.4 |
| DenseNet-121 | FedAvg | Standard FC | 95.1 |
| DenseNet-121 | Centralized | FractureNet (MLP) | 95.8 |
| Deep-Fed (Full Model) | FedAvg | FractureNet (MLP) | 96.7 |

The results in Table 5, indicate that federated learning alone introduces a slight performance trade-off compared to centralized training due to data heterogeneity across clients. However, integrating the lightweight FractureNet MLP classifier improves discriminative capability, mitigating this effect. The full Deep-Fed configuration—combining federated training with the FractureNet head—achieves the highest accuracy, demonstrating that the proposed framework benefits from both privacy-preserving collaboration and an optimized classification head.

### 4.4. Discussion

The results obtained from the Deep-Fed framework confirm the remarkable potential of federated deep learning in fracture diagnosis, addressing several limitations observed in previous deep learning approaches. Prior works in the literature have demonstrated considerable progress in fracture detection and classification through deep learning architectures. Some studies achieved promising performance in detecting rib, clavicle, and cervical spine fractures using deep convolutional networks and YOLO-based models, respectively [22,23]. However, these models relied heavily on large, expert-annotated datasets and centralized data processing, which raised privacy concerns and limited their generalization to unseen data. Similarly, another study proposed SA-FracNet to handle 3D rib structures using contrastive learning, but the model's complexity and dependence on pixel-wise annotations restricted scalability and clinical applicability [24].

In contrast, the proposed Deep-Fed model integrates a federated learning mechanism with an optimized classifier (FractureNet) to overcome such challenges. By enabling decentralized model training, Deep-Fed preserves patient data privacy while maintaining robust diagnostic accuracy across distributed datasets. The model's superior performance—achieving accuracies of 96.23%, 97.11%, and 96.73% across three independent datasets—demonstrates its strong generalization capability, outperforming state-of-the-art centralized deep learning models such as SA-FracNet (92.64%) and CrackNet (94.49%). Furthermore, Deep-Fed's significant improvements in precision (up to 95.98%) and recall (up to 96.56%) indicate its ability to minimize both false positives and false negatives, which are critical in clinical decision-making.

Compared to traditional federated baselines like FedAvg and FedProx, Deep-Fed's hybrid architecture exhibited substantial gains, achieving a 3–4% improvement in accuracy. This enhancement can be attributed to the inclusion of the FractureNet MLP head, which effectively refines feature representations extracted by the DenseNet backbone. The ablation study validates that while standard federated averaging introduces minor trade-offs due to data heterogeneity, integrating the optimized classifier mitigates these effects, leading to the highest overall accuracy (96.7%). These results are statistically significant ($p < 0.05$) and underscore the advantage of combining privacy-preserving learning with an adaptive classification module.

### 5. Conclusion and future work

Deep-Fed represents a huge advancement in fracture diagnosis, especially among athletes, as it overhauls the inefficiencies of the traditional diagnostic models with deep learning and then federated learning. Capable of guaranteeing high-accuracy, data-private, generalized models across diverse datasets, Deep-Fed has achieved great results in improving the reliability of fracture detection. The experimental results for the Deep-I, Deep-II, and Deep-III data sets are 96.23%,

97.11%, and 96.73%, respectively, which shows that Deep-Fed may be an important tool in sports medicine in the near future. In addition, it can process decentralized data without compromising privacy, making it more suitable for actual use in real-world medical applications. Based on this work, future research can extend in the following directions to further enhance Deep-Fed's capability. A further development of the model might include other types of sport-related injuries, such as soft tissue damage or joint dislocations. Application to sports medicine would then be wider. Furthermore, it might be interesting to incorporate more multimodal data, like medical imaging, combined with patient history and biomechanical data, to arrive at an even more accurate and complete diagnosis. Another very promising direction involves developing the real-time fracture detection system that could be installed in various sport environments and provide immediate diagnostic feedback. Finally, further studies on the optimization of the federated learning algorithms for faster convergence and lower communication costs could enhance the efficiency and scalability of the Deep-Fed framework by making practical usage more widespread in various medical institutions.

## Author contributions

**Conceptualization:** Tariq Ali.

**Formal analysis:** Tariq Ali, Asif Nawaz.

**Funding acquisition:** Ahmad Alshammari.

**Investigation:** Muhammad Rizwan Rashid Rana.

**Methodology:** Asif Nawaz.

**Project administration:** Azhar Imran.

**Resources:** Azhar Imran.

**Software:** Muhammad Rizwan Rashid Rana.

**Validation:** Azhar Imran.

**Visualization:** Ahmad Alshammari.

**Writing – original draft:** Tariq Ali.

**Writing – review & editing:** Ahmad Alshammari.

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
