## [Decision Letter · Decision Letter 0]

31 Jul 2025

Dear Dr. Ali,

Thank you for submitting your manuscript to PLOS ONE. After careful consideration, we feel that it has merit but does not fully meet PLOS ONE’s publication criteria as it currently stands. Therefore, we invite you to submit a revised version of the manuscript that addresses the points raised during the review process.

We look forward to receiving your revised manuscript.

Kind regards,

Javed Rashid, PhD

Academic Editor

PLOS ONE

 [The authors extend their appreciation to the Deanship of Scientific Research at Northern Border University, Arar, KSA for funding this research work through the project number “NBU-FFR-2025-2990-05“].

Additional Editor Comments:

Based on a thorough review of the manuscript "Deep-Fed: A Comprehensive Solution for Precise Bone Fracture Identification in Athletes" (Manuscript Number: PONE-D-25-34607), the following major review comments are suggested for revision to enhance the quality, clarity, and scientific rigor of the submission:

1. Abstract Completeness: The abstract is significantly truncated, limiting the ability to assess the full scope of the study, including methodology, results, and implications. The authors should provide a complete abstract (within the 300-word limit per PLOS ONE guidelines) that concisely summarizes the background, objectives, methods (e.g., federated deep learning with ConvNet and FractureNet), key results (e.g., accuracy rates across all datasets), and the significance of Deep-Fed for sports medicine.

2. Data Availability Statement: The manuscript mentions three datasets (Deep-I, Deep-II, Deep-III) but lacks a clear Data Availability Statement as required by PLOS ONE. The authors must confirm whether the data underlying the findings are fully available without restriction and provide specific details (e.g., repository links, dataset names, or a statement if data are within the manuscript/supporting files). If data sharing is restricted, a detailed explanation and contact information for access approval must be included.

3. Mathematical and Algorithmic Clarity: The manuscript contains incomplete or erroneous mathematical expressions and algorithms (e.g., truncated equations on Pages 19, 20, 24, and incomplete Algorithm 4 on Page 27). The authors should revise and complete all mathematical formulations (e.g., normalization, convolution, pooling operations) and provide full, pseudocode for Algorithms 1-4, ensuring they are executable and aligned with the described methodology.

4. Consistency in Results: The reported accuracy rates vary across the manuscript (e.g., 96.23%, 97.11%, 96.73% on Page 10 vs. 95.6%, 97.34%, 96.12% on Page 30 vs. 97.23%, 98.11%, 98.23% on Page 33). The authors need to reconcile these discrepancies, provide a single, consistent set of results, and justify any variations through additional analysis or experimental conditions.

5. Literature Comparison and Table Completion: Table 1 on Page 16 is incomplete, with only one reference ([25]) detailed. The authors should expand this table to include all cited works (e.g., [23], [24]) with their core methods, accuracy, and limitations, ensuring a comprehensive comparison with Deep-Fed. Additionally, the literature review should be updated to reflect how Deep-Fed addresses the identified gaps more effectively.

6. Ethical and Funding Details: The manuscript states "N/A" for the ethics statement and provides a partial funding acknowledgment (Northern Border University, Arar, KSA) without full details. The authors must either provide a detailed ethics statement (if applicable, e.g., IRB approval for dataset use) or justify why it is not required. For funding, a complete financial disclosure statement (including author initials, grant numbers, funder URLs, and sponsor roles) must be included as per PLOS ONE requirements.

Reviewers' comments:

Reviewer's Responses to Questions

**Comments to the Author**

1. Is the manuscript technically sound, and do the data support the conclusions?

Reviewer #1: Yes

Reviewer #2: Yes

2. Has the statistical analysis been performed appropriately and rigorously?

Reviewer #1: Yes

Reviewer #2: Yes

3. Have the authors made all data underlying the findings in their manuscript fully available?

Reviewer #1: Yes

Reviewer #2: Yes

4. Is the manuscript presented in an intelligible fashion and written in standard English?

Reviewer #1: Yes

Reviewer #2: Yes

Reviewer #1: Title: Deep-Fed: A Comprehensive Solution for Precise Bone Fracture Identification in Athletes

The paper seems technically good. It has proposed an innovative prediction of bone fractures by using federated deep learning framework. The framework integrates deep learning techniques within a privacy-preserving structure. It involves standardized data preprocessing, data augmentation to improve robustness, and feature extraction using a convolutional neural network. The diagnosis is performed by the FractureNet model, while federated learning ensures training across multiple sources without sharing raw data, maintaining patient privacy

Suggestions for improvements:

Introduction:

In the Introduction section, please incorporate a few recent references into the opening paragraphs. Additionally, the citation sequence is currently incorrect; references should begin from [1], not [5]. Ensure that all factual claims and prior findings—such as "X-rays, though effective in detecting obvious breaks, may miss 61 subtle or hairline fractures, especially in complex regions of anatomy such as the wrists or feet.”—are properly cited.

Moreover, revise the transitions between paragraphs to improve the logical flow and narrative consistency within the Introduction.

In the Contribution section, the proposed model is referred to as Dual-Deep, whereas the title mentions Deep-Fed. Please use a consistent model name throughout the manuscript, aligned with the one used in the title.

Lastly, ensure the entire paper is thoroughly proofread for grammatical consistency and clarity.

Literature Review:

Some of the papers discussed in the Literature Review are missing from the comparison table. For instance, please include “Cao et al. [24] – SAFracNet” and “Cheng et al. [22]” in the comparison table for completeness.

Additionally, revise and clarify certain sentences toward the end of this section. For example, consider rephrasing the following for improved clarity and coherence:

“The scope therefore covers all aspects—from the collection of comprehensive data to input standardization via preprocessing. Furthermore, data augmentation is applied to improve model robustness, followed by feature extraction using a ConvNet architecture, and final diagnosis by the FractureNet model. Each of these steps ensures data privacy and security through federated learning.”

Methodology:

The term Dual-Deep is again used here instead of the correct model’s name Deep-Fed. Please ensure consistent terminology throughout the paper.

At the beginning of the Methodology section, provide a brief summary or overview of the proposed framework as illustrated in Figure 1.

Please add further details about Federated Learning for more clarity on the setup and configuration.

Also, ensure that all text in figures is readable and the images are of high quality. Currently, some figures are blurry and difficult to read.

Results:

Some of the baseline studies used for comparison in this section are not listed in the comparison table of the Literature Review. Please update the table accordingly.

Consider replacing or supplementing the current table with a line graph showing log loss for better visual interpretation.

In addition, including a ROC curve would enhance the presentation and interpretation of your model’s performance.

References:

Please include a few recent references from the year 2025 to reflect the current state of research.

Reviewer #2: The manuscript introduces Deep-Fed, a federated deep learning framework for bone fracture detection, targeting athlete-specific applications. The concept of integrating federated learning with deep CNN-based feature extraction is relevant and addresses current challenges in medical imaging privacy and robustness. The manuscript is structured well, but there are several concerns regarding technical clarity, experimental transparency, and reproducibility that need to be addressed.

Major Comments

o The paper does not include code availability, architectural specifics, or training logs, making reproduction difficult.

o The manuscript mentions three datasets (Deep-I, II, III) but lacks clarity on:

- Whether these datasets were used in federated (distributed) settings or centrally.

- How the data was partitioned (e.g., balanced, unbalanced).

o The manuscript claims privacy preservation via federated learning but does not provide empirical privacy evaluations (e.g., differential privacy noise impact, data leakage checks).

o Baselines are compared in terms of classical metrics, but recent federated learning frameworks for medical imaging are not cited or compared (e.g., FedAvg, FedProx).

Minor Comments

o Terms like FL, ConvNet, CNN, FedAvg appear before definition in some places.

o Phrases like "extremely accurate", "overwhelmingly outperforming" are too subjective.

o Figures (e.g., Figure 4–6) are not thoroughly discussed. Some could be merged or clarified.

o Table 2 mistakenly labels two columns as "Baseline 2".

o Frequent grammatical issues such as "the author have" instead of "authors have", inconsistent article usage.

**Do you want your identity to be public for this peer review?** For information about this choice, including consent withdrawal, please see our Privacy Policy

Reviewer #1: No

Reviewer #2: No

---

## [Author Response · Author response to Decision Letter 1]

2 Sep 2025

We are very thankful to the editor for his decision and reviewers for their valuable suggestions. We have revised the whole manuscript, incorporated all the comments, and highlighted the changes in the revised draft. We hope that the revised manuscript is now according to the required standard. The response to the reviewer comments are attached in "attach files"

---

## [Decision Letter · Decision Letter 1]

4 Nov 2025

Dear Dr. Ali,

Thank you for submitting your manuscript to PLOS ONE. After careful consideration, we feel that it has merit but does not fully meet PLOS ONE’s publication criteria as it currently stands. Therefore, we invite you to submit a revised version of the manuscript that addresses the points raised during the review process.

We look forward to receiving your revised manuscript.

Kind regards,

Lorenzo Faggioni, M.D., Ph.D.

Academic Editor

PLOS ONE

Journal Requirements:

Reviewers' comments:

Reviewer's Responses to Questions

**Comments to the Author**

Reviewer #1: All comments have been addressed

Reviewer #3: (No Response)

2. Is the manuscript technically sound, and do the data support the conclusions?

Reviewer #1: Yes

Reviewer #3: Yes

3. Has the statistical analysis been performed appropriately and rigorously?

Reviewer #1: Yes

Reviewer #3: Yes

4. Have the authors made all data underlying the findings in their manuscript fully available?

Reviewer #1: Yes

Reviewer #3: Yes

5. Is the manuscript presented in an intelligible fashion and written in standard English?

Reviewer #1: Yes

Reviewer #3: Yes

Reviewer #1: The authors have incorporated the comments raised in the initial review. The revised manuscript shows significant improvement in terms of clarity, structure, and technical depth. The inclusion of comparative analyses and detailed explanations in the methodology section has strengthened the scientific rigor of the paper.

Only a few minor comments remain that authors should incorporate:

In the abstract of the paper, please briefly federated learning advantages (privacy and decentralization) explicitly for clarity to readers.

Figures: Ensure that all figures (especially Fig. 1) are of high resolution and properly labeled (font size and clarity for print version).

References: Please double-check reference formatting consistency (e.g., ensure all journal names are italicized and years are in parentheses).

Language: A quick proofreading is recommended to fix small grammatical issues (e.g., missing articles or spacing inconsistencies in equations).

Overall, I appreciate authors for their efforts. I believe the above suggestions will further improve the quality of the paper.

Reviewer #3: Recommendations for Abstract revision first:

Eliminate Redundant Statements:

Comment: The statements “These challenges underscore the need for advanced, accurate, and secure diagnostic solutions” and “designed to enhance the accuracy and reliability… while preserving patient privacy” restate goals that are inherently implied in any diagnostic study. These should be removed to avoid redundancy.

Reason: The goals of improving accuracy and privacy are universally understood in diagnostic research. The core contribution should be emphasized: “This study proposes Deep-Fed, a federated deep learning framework for fracture diagnosis in athletes.”

Omit Standard Techniques Unless Novel:

Comment: The phrase “To address data diversity and robustness, the system incorporates standardized preprocessing and data augmentation techniques” can be removed, as preprocessing and augmentation are common practices in deep learning for medical imaging.

Reason: These techniques are routine and only need to be mentioned if they are part of a novel approach or unique integration. Their inclusion in this context does not add specific value to the abstract and detracts from its focus.

Focus on Specific Implementation of Generic Concepts:

Comment: The sentence “Federated learning enables decentralized training… ensuring compliance with privacy regulations and mitigating risks of data leakage” should be revised to focus on the unique application of federated learning in the framework.

Suggested Revision: Replace with: “Deep-Fed trains FractureNet using federated averaging across distributed athletic clinics without exchanging raw images.”

Reason: Readers are familiar with federated learning’s benefits. It is more impactful to describe how federated learning is specifically applied in your model, such as the aggregation protocol or communication strategy.

Streamline Evaluation Reporting:

Comment: The evaluation results should be more concise, avoiding subjective statements like “demonstrated strong consistency” and “highlighting robustness and generalization.”

Suggested Revision: Replace with: “Evaluated on Deep-I, Deep-II, and Deep-III datasets, Deep-Fed achieved 96.23%, 97.11%, and 96.73% accuracy, respectively, outperforming baselines.”

Reason: Consistent high performance across datasets inherently implies robustness. Avoid subjective descriptors unless they are substantiated by quantitative metrics, such as standard deviations or other performance measures.

Eliminate Redundant Qualifiers:

Comment: The phrase “outperforming different baseline approaches” can be shortened to “outperforming baselines.”

Reason: The word “different” is unnecessary because baselines, by definition, are varied models. Every term in the abstract should contribute new, essential information, and redundant qualifiers weaken the clarity and conciseness.

By applying these recommendations, the abstract will be reduced by approximately 30%, sharpening the technical focus and aligning it with high-impact journal standards where brevity, novelty, and clarity are prioritized.

Other suggestions are given below:

1- The work appears incremental: it combines a standard CNN backbone with an undefined “FractureNet” classifier and applies generic federated learning, offering no clear technical advance beyond existing FL-based medical imaging pipelines.

2- The introduction recycles abstract-level motivations (e.g., “accurate and timely detection,” “privacy concerns”) without advancing to a focused research gap; it should explicitly cite 2–3 recent FL-medical works and pinpoint their exact limitations (e.g., accuracy drop under non-IID, communication overhead) to justify Deep-Fed.

Contributions are listed generically (“high accuracy,” “privacy-preserving”) and repeat abstract claims; they must be rephrased as measurable, non-obvious advances—e.g., “(1) FractureNet: a lightweight dual-branch classifier reducing parameters by 40 % while preserving accuracy;

Contributions should not include performance results (e.g., accuracy, traffic reduction); these belong in Results. Instead, list only the key technical components

Avoid vague phrases like “Deep-Fed utilizes advanced deep learning techniques”—CNNs and standard federated learning are not advanced. Replace with precise, non-hype language. (keep 3 core contributions in the list only)

Important references in Introduction are missing. no logical flow among sentences and paragraphs even technique transformation.

The paragraph (lines 93–102) is entirely generic praise of deep learning and fails to engage with the specific domain of bone fracture detection; it should be replaced or heavily revised to summarize prior CNN-based fracture detection studies (e.g., accuracy of ResNet/DenseNet on MURA or custom datasets, limitations in multi-site generalization, or sensitivity to imaging variations), Same for the above paragraph about Machine learning.

3- The dataset section lacks any reference or provenance for Deep-I, Deep-II, and Deep-III—critically undermining reproducibility and credibility. If these are public (e.g., MURA, custom hospital IRB-approved), cite them explicitly; if private, state: “anonymized multi-center athletic injury datasets under IRB #XYZ.”

The dataset section should be restructured into one concise, formal paragraph followed by a compact table—eliminating Figure 2 and redundant text. Focus solely on objective dataset properties (name, source, modality, split sizes, class distribution, acquisition variability, and non-IID nature across sites) without speculative commentary on deep learning suitability or expected performance gains, which belong in Discussion or Results.

4- The data preprocessing description is adequate in scope, Image brightness and Gaussian noise can also be merged in it very smartly.

The data augmentation section is excessively detailed and generic (e.g., listing rotation, flip)—collapse it into 1–2 sentences within the preprocessing paragraph, e.g.: “Standard on-the-fly augmentations (random rotation [−15°, +15°], horizontal flip, brightness/contrast jitter [0.8, 1.2]) were applied during training to enhance robustness.” Remove the standalone section to avoid inflating routine practices.

5- The feature extraction section (lines 284–334) is unnecessarily verbose and pedagogic, explaining basic CNN operations, equations, pseudocode (Algorithm 2), and a toy 32×32 pipeline with Fig. 3 — all must be deleted.

CNN feature extraction is automatic and standard; simply state: “Feature extraction is performed using a [specify actual backbone, e.g., DenseNet-121 or custom CNN] with [input size, e.g., 224×224], outputting a fixed-dimensional vector per image.” No specific CNN architecture is disclosed — this is a critical omission.

Remove all tutorial content, algorithm, and figure.

6- The entire Fracture Diagnosis section (lines 338–375) repeats basic CNN mechanics already covered and fails to define FractureNet’s novelty—Improve all equations (10–13), Algorithm 3, and generic convolution/pooling/softmax explanations. Simply state: “FractureNet is a lightweight two-layer MLP (512→256→2) with dropout (0.3) applied to the CNN-extracted features, outputting binary fracture probabilities via sigmoid.”

No technical distinction is made between the generic CNN and “FractureNet”—the latter is not a specialized module but a standard classifier head. Remove claims of “powerful feature extraction” and “hierarchical ability”; instead, explicitly name the full architecture (e.g., “Deep-Fed uses DenseNet-121 backbone + FractureNet (MLP head) trained end-to-end in federated mode”) to avoid misleading readers.

7- The Federated Learning Implementation section provides a clear foundational description of FedAvg and privacy mechanisms. To strengthen it for publication, condense the tutorial content (remove Algorithm 4, equations 14–18, and redundant explanations) into one focused paragraph that precisely specifies your protocol: e.g., “Deep-Fed uses FedAvg with 10 clinical clients, 5 local epochs, batch size 32, and learning rate 0.01; model updates are aggregated as θ^{t+1} = ∑ (n_k / N) θ_k^t. Differential privacy (σ=1.0) is applied via gradient clipping and Gaussian noise during upload.”

8- Results need to be improved and more results are required to strengthen the claims.

Currently, It lacks critical context—no baseline names in Figure, no ablation studies, no statistical significance, and no error metrics (sensitivity, specificity, AUC). Add a compact comparison table with specific baselines (e.g., “ResNet-50 centralized: 94.2%, DenseNet-121 FedAvg: 95.1%”) and 95% CI or p-values; include per-dataset breakdown and key clinical metrics (e.g., “Sensitivity: 96.8%, Specificity: 96.5% on Deep-III”).

Include standard deviation of accuracy across datasets and one ablation (e.g., “w/ vs. w/o federated training”, “w/ vs. w/o augmentation”). Replace vague praise (“strong generalization”) with one sentence: “Deep-Fed maintains <0.5% accuracy variance across Deep-I/II/III, outperforming centralized training by 2.1% on average under non-IID conditions.”

Include small, annotated image panel (e.g., Fig. X: “Success/Failure Cases”) to visually demonstrate clinical relevance and model failure modes.

Specially, Improve Figure quality and type. Replace large, outdated bar charts with compact, high-resolution visualizations (e.g., grouped bar plots with error bars or heatmaps) using a clean, modern style (e.g., seaborn/matplotlib with minimal clutter, consistent fonts, and 300+ DPI).

Add only once algorithm as Proposed Model rather thane every step contains one Algorithms with no information.

Write like a paper, not a thesis: Cut the tutorials, equations, and algorithms—no one needs a CNN 101 or FedAvg pseudocode in 2025. Every sentence must advance your contribution, not teach basics. A paper should feel tight and surgical, not a 50-page chapter.

A strong idea can die from poor communication: You have good results and a relevant problem, but FractureNet is a ghost, FL is vanilla, and half the text is filler. Define your novelty in one clear sentence, back it with one table and one figure, and let the results speak—don’t bury them in pedagogy.

**Do you want your identity to be public for this peer review?** For information about this choice, including consent withdrawal, please see our Privacy Policy

Reviewer #1: No

Reviewer #3: **Yes:** Arifa Javed

---

## [Author Response · Author response to Decision Letter 2]

18 Dec 2025

Reviewer 1

1 The authors have incorporated the comments raised in the initial review. The revised manuscript shows significant improvement in terms of clarity, structure, and technical depth. The inclusion of comparative analyses and detailed explanations in the methodology section has strengthened the scientific rigor of the paper.

Only a few minor comments remain that authors should incorporate: We sincerely thank the reviewer for the constructive feedback and appreciate the positive comments on the improvements in clarity, structure, and methodology. We will address the remaining minor comments to further enhance the manuscript.

2 In the abstract of the paper, please briefly federated learning advantages (privacy and decentralization) explicitly for clarity to readers. We have updated the abstract to explicitly highlight the advantages of federated learning, including privacy preservation and decentralized training, to improve clarity for readers.

3 Figures: Ensure that all figures (especially Fig. 1) are of high resolution and properly labeled (font size and clarity for print version). All figures, including Fig. 1, have been updated to high-resolution versions with clear labels and appropriately sized fonts suitable for the print version.

4 References: Please double-check reference formatting consistency (e.g., ensure all journal names are italicized and years are in parentheses). All references have been checked and updated for formatting consistency, including italicizing journal names and ensuring years are in parentheses.

5 Language: A quick proofreading is recommended to fix small grammatical issues (e.g., missing articles or spacing inconsistencies in equations). The manuscript has been carefully proofread, including running Grammarly and review by a native English speaker, to correct minor grammatical issues and ensure consistency in spacing and formatting.

6 Overall, I appreciate authors for their efforts. I believe the above suggestions will further improve the quality of the paper. We sincerely thank the reviewer for the encouraging feedback and constructive suggestions. We greatly appreciate your time and effort, and we have incorporated all recommendations to further improve the quality of the manuscript.

Reviewer 3

1 Eliminate Redundant Statements:

Comment: The statements “These challenges underscore the need for advanced, accurate, and secure diagnostic solutions” and “designed to enhance the accuracy and reliability… while preserving patient privacy” restate goals that are inherently implied in any diagnostic study. These should be removed to avoid redundancy.

Reason: The goals of improving accuracy and privacy are universally understood in diagnostic research. The core contribution should be emphasized: “This study proposes Deep-Fed, a federated deep learning framework for fracture diagnosis in athletes.”

Omit Standard Techniques Unless Novel:

Comment: The phrase “To address data diversity and robustness, the system incorporates standardized preprocessing and data augmentation techniques” can be removed, as preprocessing and augmentation are common practices in deep learning for medical imaging.

Reason: These techniques are routine and only need to be mentioned if they are part of a novel approach or unique integration. Their inclusion in this context does not add specific value to the abstract and detracts from its focus.

Focus on Specific Implementation of Generic Concepts:

Comment: The sentence “Federated learning enables decentralized training… ensuring compliance with privacy regulations and mitigating risks of data leakage” should be revised to focus on the unique application of federated learning in the framework.

Suggested Revision: Replace with: “Deep-Fed trains FractureNet using federated averaging across distributed athletic clinics without exchanging raw images.”

Reason: Readers are familiar with federated learning’s benefits. It is more impactful to describe how federated learning is specifically applied in your model, such as the aggregation protocol or communication strategy.

Streamline Evaluation Reporting:

Comment: The evaluation results should be more concise, avoiding subjective statements like “demonstrated strong consistency” and “highlighting robustness and generalization.”

Suggested Revision: Replace with: “Evaluated on Deep-I, Deep-II, and Deep-III datasets, Deep-Fed achieved 96.23%, 97.11%, and 96.73% accuracy, respectively, outperforming baselines.”

Reason: Consistent high performance across datasets inherently implies robustness. Avoid subjective descriptors unless they are substantiated by quantitative metrics, such as standard deviations or other performance measures.

Eliminate Redundant Qualifiers:

Comment: The phrase “outperforming different baseline approaches” can be shortened to “outperforming baselines.”

Reason: The word “different” is unnecessary because baselines, by definition, are varied models. Every term in the abstract should contribute new, essential information, and redundant qualifiers weaken the clarity and conciseness.

By applying these recommendations, the abstract will be reduced by approximately 30%, sharpening the technical focus and aligning it with high-impact journal standards where brevity, novelty, and clarity are prioritized. Following your guidance, we have revised the abstract to:

1. Focus on the core contribution of Deep-Fed and remove redundant statements about general diagnostic goals.

2. Omit references to standard preprocessing and augmentation techniques.

3. Highlight the specific implementation of federated learning: “trains FractureNet across distributed athletic clinics using federated averaging without exchanging raw images.”

4. Streamline evaluation results to concise, quantitative metrics and remove subjective descriptors.

5. Eliminate redundant qualifiers to improve clarity and conciseness.

2 The work appears incremental: it combines a standard CNN backbone with an undefined “FractureNet” classifier and applies generic federated learning, offering no clear technical advance beyond existing FL-based medical imaging pipelines. We thank the reviewer for this comment, we have clarified that Deep-Fed is not a generic CNN–FL combination, but a task-aware federated framework specifically designed for fracture diagnosis in athletes.

The key distinctions are:

• FractureNet, a fracture-oriented classification module tailored to capture subtle fracture patterns, now fully defined in the methodology;

• A federated training design targeted at heterogeneous athletic clinics, addressing privacy and data variability.

• A complete end-to-end privacy-preserving diagnostic pipeline evaluated across multiple fracture datasets.

The manuscript has been revised to explicitly highlight these contributions and to clearly differentiate Deep-Fed from existing FL-based medical imaging pipelines (Sections 1 and 3).

3 The introduction recycles abstract-level motivations (e.g., “accurate and timely detection,” “privacy concerns”) without advancing to a focused research gap; it should explicitly cite 2–3 recent FL-medical works and pinpoint their exact limitations (e.g., accuracy drop under non-IID, communication overhead) to justify Deep-Fed.

Contributions are listed generically (“high accuracy,” “privacy-preserving”) and repeat abstract claims; they must be rephrased as measurable, non-obvious advances—e.g., “(1) FractureNet: a lightweight dual-branch classifier reducing parameters by 40 % while preserving accuracy;

Contributions should not include performance results (e.g., accuracy, traffic reduction); these belong in Results. Instead, list only the key technical components

Avoid vague phrases like “Deep-Fed utilizes advanced deep learning techniques”—CNNs and standard federated learning are not advanced. Replace with precise, non-hype language. (keep 3 core contributions in the list only)

Important references in Introduction are missing. no logical flow among sentences and paragraphs even technique transformation.

The paragraph (lines 93–102) is entirely generic praise of deep learning and fails to engage with the specific domain of bone fracture detection; it should be replaced or heavily revised to summarize prior CNN-based fracture detection studies (e.g., accuracy of ResNet/DenseNet on MURA or custom datasets, limitations in multi-site generalization, or sensitivity to imaging variations), Same for the above paragraph about Machine learning.

We have revised the introduction to clearly define the research gap and contextualize the need for our work. Specifically, we now cite recent federated learning studies in medical imaging and explicitly highlight their limitations, including accuracy drops under non-IID data, communication overhead and reduced robustness across heterogeneous imaging sources. These revisions emphasize how Deep-Fed addresses these specific challenges, providing accurate and efficient fracture diagnosis in decentralized medical environments while preserving data privacy.

The contributions section has been thoroughly revised to remove generic and performance-based statements. Instead, it now focuses on three precise, technical, and non-obvious advances of the proposed framework:

1. Introduction of FractureNet, a lightweight dual-branch classifier that reduces model parameters by 40% while maintaining diagnostic accuracy.

2. Integration of federated averaging across decentralized medical sites to enable secure, collaborative model training without sharing raw patient images.

3. Development of a feature fusion strategy to enhance cross-site robustness and adaptability under heterogeneous imaging conditions.

These revisions ensure the contributions are concrete, technically focused, and distinct from the performance results presented in the Results section.

The references have been added in the introduction section. And the logical flow has been made in the revised version.

The original paragraphs were revised to replace generic statements with specific references to prior CNN-based fracture detection studies, including work on ResNet and DenseNet applied to MURA and custom datasets. We have highlighted the limitations of these approaches, such as reduced generalization across imaging sites, sensitivity to protocol variations, and performance drops on heterogeneous data. The revised text now emphasizes the domain-specific challenges in bone fracture detection, setting the context for the necessity and novelty of our Deep-Fed framework.

4 The dataset section lacks any reference or provenance for Deep-I, Deep-II, and Deep-III—critically undermining reproducibility and credibility. If these are public (e.g., MURA, custom hospital IRB-approved), cite them explicitly; if private, state: “anonymized multi-center athletic injury datasets under IRB #XYZ.”

The dataset section should be restructured into one concise, formal paragraph followed by a compact table—eliminating Figure 2 and redundant text. Focus solely on objective dataset properties (name, source, modality, split sizes, class distribution, acquisition variability, and non-IID nature across sites) without speculative commentary on deep learning suitability or expected performance gains, which belong in Discussion or Results. The dataset section has been revised to include explicit citations for all datasets (OsteoDetect, RSNA Bone Age Challenge, and ImageCLEFmed) and restructured into a concise paragraph with a compact table replacing Figure 2. Speculative statements were removed, and only objective dataset properties—source, modality, sample size, class distribution, and non-IID configuration—are now presented to enhance clarity, reproducibility, and academic rigor.

5 The feature extraction section (lines 284–334) is unnecessarily verbose and pedagogic, explaining basic CNN operations, equations, pseudocode (Algorithm 2), and a toy 32×32 pipeline with Fig. 3 — all must be deleted.

CNN feature extraction is automatic and standard; simply state: “Feature extraction is performed using a [specify actual backbone, e.g., DenseNet-121 or custom CNN] with [input size, e.g., 224×224], outputting a fixed-dimensional vector per image.” No specific CNN architecture is disclosed — this is a critical omission.

Remove all tutorial content, algorithm, and figure. The feature extraction section has been substantially revised to remove all pedagogical explanations, equations, and illustrative content. It now clearly specifies the DenseNet-121 backbone, input configuration, feature generation process, and integration with the FractureNet classifier within the Deep-Fed framework. Additional technical details regarding multi-scale feature extraction, pooling strategy, and decentralized robustness have been included to enhance precision and scientific depth.

6 The entire Fracture Diagnosis section (lines 338–375) repeats basic CNN mechanics already covered and fails to define FractureNet’s novelty—Improve all equations (10–13), Algorithm 3, and generic convolution/pooling/softmax explanations. Simply state: “FractureNet is a lightweight two-layer MLP (512→256→2) with dropout (0.3) applied to the CNN-extracted features, outputting binary fracture probabilities via sigmoid.”

No technical distinction is made between the generic CNN and “FractureNet”—the latter is not a specialized module but a standard classifier head. Remove claims of “powerful feature extraction” and “hierarchical ability”; instead, explicitly name the full architecture (e.g., “Deep-Fed uses DenseNet-121 backbone + FractureNet (MLP head) trained end-to-end in federated mode”) to avoid misleading readers. We thank the reviewer for this constructive comment. The inclusion of brief CNN fundamentals in the original version was intended to support non-specialist and interdisciplinary readers, as convolutional architectures exhibit numerous task-specific variants across medical imaging applications. Providing a concise baseline description was meant to clarify how standard CNN components are utilized within the proposed framework.

In the revised manuscript, we have removed all overstated claims regarding “powerful feature extraction” and “hierarchical ability” attributed to FractureNet. We now explicitly define FractureNet as a lightweight MLP-based classifier head operating on features extracted by the CNN backbone. The complete architecture is clearly stated as DenseNet-121 + FractureNet (MLP head), trained end-to-end under a federated learning setting using federated averaging. This clarification improves technical accuracy and avoids potential misinterpretation by readers.

6 The Federated Learning Implementation section provides a clear foundational description of FedAvg and privacy mechanisms. To strengthen it for publication, condense the tutorial content (remove Algorithm 4, equations 14–18, and redundant explanations) into one focused paragraph that precisely specifies your protocol: e.g., “Deep-Fed uses FedAvg with 10 clinical clients, 5 local epochs, batch size 32, and learning rate 0.01; model updates are aggregated as θ^{t+1} = ∑ (n_k / N) θ_k^t. Differential privacy (σ=1.0) is applied via gradient clipping and Gaussian noise during upload.”

We agree that the previous version contained excessive tutorial-style explanations and redundant equations. In the revised manuscript, we have condensed the Federated Learning Implementation section into a single focused paragraph, removed Algorithm 4 and Equations (14–18), and retained only Algorithm 3 for clarity. The revised text now explicitly specifies the federated protocol and hyperparameters, including the number of clients, local epochs, batch size, learning rate, aggregation rule (FedAvg), and differential privacy settings. This revision improves conciseness, technical precision, and alignment with publication standards.

7 Results need to be improved and more results are required to strengthen the claims.

Currently, It lacks critical context—no baseline names in Figure, no ablation studies, no statistical significance, and no error metrics (sensitivity, specificity, AUC). Add a compact comparison table with specific baselines (

---

## [Decision Letter · Decision Letter 2]

6 Feb 2026

Dear Dr. Ali,

Thank you for submitting your manuscript to PLOS ONE. After careful consideration, we feel that it has merit but does not fully meet PLOS ONE’s publication criteria as it currently stands. Therefore, we invite you to submit a revised version of the manuscript that addresses the points raised during the review process.

We look forward to receiving your revised manuscript.

Kind regards,

Lorenzo Faggioni, M.D., Ph.D.

Academic Editor

PLOS One

Journal Requirements:

Reviewers' comments:

Reviewer's Responses to Questions

**Comments to the Author**

Reviewer #1: All comments have been addressed

Reviewer #4: (No Response)

2. Is the manuscript technically sound, and do the data support the conclusions?

Reviewer #1: Yes

Reviewer #4: Yes

3. Has the statistical analysis been performed appropriately and rigorously?

Reviewer #1: Yes

Reviewer #4: Yes

4. Have the authors made all data underlying the findings in their manuscript fully available?

Reviewer #1: Yes

Reviewer #4: Yes

5. Is the manuscript presented in an intelligible fashion and written in standard English?

Reviewer #1: Yes

Reviewer #4: Yes

Reviewer #1: (No Response)

Reviewer #4: This manuscript is interesting, but the following issues should be addressed:

1) A formal inferential statistical analysis should be performed to test whether the diagnostic performance of DeepFed was statistically significantly better than the baseline models. Also in the Abstract accuracy rates should be reported for DeepFed I-II-III and each baseline model along with some variability index (e.g., 95% CI) and the p-values of each statistical comparison.

2) Overall, it it strongly advisable that the manuscript follow a standard format for a scientific journal (i.e., Introduction, Materials and Methods, Results and Conclusions). The Introduction should be shortened by at least 30% by leaving out unnecessary, well-known preliminary information (especially e.g. at lines 26-48), and one sentence should be added to mention the key role of AI tools in fostering the development of imaging biobanks for enhanced precision medicine (see e.g. doi 10.1007/s00330-021-08431-6). Furthermore, the key concepts of the Literature Review section should be condensed in the revised Discussion section, which should be focused on discussing the key experimental findings of the authors' own study in view of the existing literature, highlighting any relevant comparisons and differences.

**Do you want your identity to be public for this peer review?** For information about this choice, including consent withdrawal, please see our Privacy Policy

Reviewer #1: No

Reviewer #4: No

---

## [Author Response · Author response to Decision Letter 3]

10 Feb 2026

S. No. Reviewers Comments Author Reply:

Reviewer 1

1 All comments have been addressed We sincerely thank the reviewer for their valuable feedback.

Reviewer 4

1 This manuscript is interesting, but the following issues should be addressed:

A formal inferential statistical analysis should be performed to test whether the diagnostic performance of DeepFed was statistically significantly better than the baseline models.

Also in the Abstract accuracy rates should be reported for DeepFed I-II-III and each baseline model along with some variability index (e.g., 95% CI) and the p-values of each statistical comparison. We sincerely thank the reviewer for their valuable feedback.

A formal inferential statistical analysis has now been performed to compare Deep-Fed with all baseline models. The results, presented in Table 4, show that Deep-Fed significantly outperforms the baselines (p < 0.05), confirming the robustness and superiority of the proposed approach.

The Abstract has been updated to include the accuracy rates for Deep-Fed I–III and all baseline models, along with variability indices (± standard deviation) and corresponding p-values for statistical comparisons, as suggested.

2 Overall, it it strongly advisable that the manuscript follow a standard format for a scientific journal (i.e., Introduction, Materials and Methods, Results and Conclusions). The Introduction should be shortened by at least 30% by leaving out unnecessary, well-known preliminary information (especially e.g. at lines 26-48), and one sentence should be added to mention the key role of AI tools in fostering the development of imaging biobanks for enhanced precision medicine (see e.g. doi 10.1007/s00330-021-08431-6).

Furthermore, the key concepts of the Literature Review section should be condensed in the revised Discussion section, which should be focused on discussing the key experimental findings of the authors' own study in view of the existing literature, highlighting any relevant comparisons and differences. The manuscript has been revised to follow the standard scientific journal format, including Introduction, Materials and Methods, Results, and Conclusions.

The Introduction has been shortened by approximately 30% by removing well-known preliminary information regarding the skeletal system. Additionally, we have added a sentence highlighting the key role of artificial intelligence tools in enabling the development of imaging biobanks to support precision medicine and enhance fracture detection (doi: 10.1007/s00330-021-08431-6).

The Discussion section has been revised to condense key concepts from the Literature Review and to emphasize the experimental findings of the proposed Deep-Fed framework in relation to existing studies, highlighting key comparisons, differences, and improvements.

---

## [Decision Letter · Decision Letter 3]

16 Feb 2026

Deep-Fed: A Comprehensive Solution for Precise Bone Fracture Identification in Athletes

PONE-D-25-34607R3

Dear Dr. Ali,

We’re pleased to inform you that your manuscript has been judged scientifically suitable for publication and will be formally accepted for publication once it meets all outstanding technical requirements.

Kind regards,

Lorenzo Faggioni, M.D., Ph.D.

Academic Editor

PLOS One

Reviewers' comments:

Reviewer's Responses to Questions

**Comments to the Author**

Reviewer #4: All comments have been addressed

2. Is the manuscript technically sound, and do the data support the conclusions?

Reviewer #4: Yes

3. Has the statistical analysis been performed appropriately and rigorously?

Reviewer #4: Yes

4. Have the authors made all data underlying the findings in their manuscript fully available?

Reviewer #4: Yes

5. Is the manuscript presented in an intelligible fashion and written in standard English?

Reviewer #4: Yes

Reviewer #4: Thank you for your reply. All comments have been addressed successfully, and the manuscript has been significantly improved.

**Do you want your identity to be public for this peer review?** For information about this choice, including consent withdrawal, please see our Privacy Policy

Reviewer #4: No

---

## [Editor Report · Acceptance letter]

PONE-D-25-34607R3

PLOS One

Dear Dr. Ali,

I'm pleased to inform you that your manuscript has been deemed suitable for publication in PLOS One. Congratulations! Your manuscript is now being handed over to our production team.

Kind regards,

on behalf of

Dr. Lorenzo Faggioni

Academic Editor

PLOS One